# Graph Neural Networks for Temporal Graphs: State of the Art, Open Challenges, and Opportunities

**Antonio Longa**[*]                                                          *antonio.longa@unitn.it*
*University of Trento and Fondazione Bruno Kessler, Trento, Italy*

**Veronica Lachi**[*]                                                        *veronica.lachi@student.unisi.it*
*University of Siena, Siena, Italy*

**Gabriele Santin**[*]                                                              *gsantin@fbk.eu*
*Fondazione Bruno Kessler, Trento, Italy*

**Monica Bianchini**                                                          *monica@diism.unisi.it*
*University of Siena, Siena, Italy*

**Bruno Lepri**                                                                      *lepri@fbk.eu*
*Fondazione Bruno Kessler, Trento, Italy*

**Pietro Liò**                                                                      *pl219@cam.ac.uk*
*University of Cambridge, Cambridge, United Kingdom*

**Franco Scarselli**                                                          *franco@diism.unisi.it*
*University of Siena, Siena, Italy*

**Andrea Passerini**                                                        *andrea.passerini@unitn.it*
*University of Trento, Trento, Italy*

**Reviewed on OpenReview:** *https://openreview.net/forum?id=pHCdMat0gI*

## Abstract

Graph Neural Networks (GNNs) have become the leading paradigm for learning on (static) graph-structured data. However, many real-world systems are dynamic in nature, since the graph and node/edge attributes change over time. In recent years, GNN-based models for temporal graphs have emerged as a promising area of research to extend the capabilities of GNNs. In this work, we provide the first comprehensive overview of the current state-of-the-art of temporal GNN, introducing a rigorous formalization of learning settings and tasks and a novel taxonomy categorizing existing approaches in terms of how the temporal aspect is represented and processed. We conclude the survey with a discussion of the most relevant open challenges for the field, from both research and application perspectives.

---

[*]*Equal contribution*

# 1 Introduction

The ability to process temporal graphs is becoming increasingly important in a variety of fields such as recommendation systems (Gao et al., 2022a; Wu et al., 2022), social network analysis (Deng et al., 2019; Fan et al., 2019), transportation systems (Jiang & Luo, 2022; Yu et al., 2017), modeling of face-to-face interactions (Longa et al., 2022c), human mobility (Mauro et al., 2022; Gao, 2015), epidemic modeling and contact tracing (Cencetti et al., 2021; So et al., 2020), and many others. Traditional graph-based models are not well suited for analyzing temporal graphs as they assume a fixed structure and are unable to capture its temporal evolution. Therefore, in the last few years, several models capable to directly encode temporal graphs have been developed, such as matrix factorization-based approaches (Ahmed et al., 2018) and temporal motif-based methods (Longa et al., 2022b). Recently, also GNNs have been successfully applied to temporal graphs. Indeed, their success in various static graph tasks, including node classification (Hamilton et al., 2017; Veličković et al., 2017; Kipf & Welling, 2016a; Gao et al., 2018; Gasteiger et al., 2018; Monti et al., 2017; Wu et al., 2019) and link prediction (Zhang & Chen, 2018; Cai & Ji, 2020; Zhang & Chen, 2017), has not only established them as the leading paradigm in static graph processing, but has also indicated the importance of exploring their potential in other graph domains, such as temporal graphs. With approaches ranging from attention-based methods (Xu et al., 2020) to Variational Graph-Autoencoders (VGAEs) (Hajiramezanali et al., 2019), Temporal Graph Neural Networks (TGNNs) have achieved state-of-the-art results on tasks such as temporal link prediction (Sankar et al., 2020), node classification (Rossi et al., 2020) and edge classification (Wang et al., 2021a). Despite the potential of GNN-based models for temporal graph processing and the variety of different approaches that emerged, a systematization of the literature is still missing. Existing surveys either discuss general techniques for learning over temporal graphs, only briefly mentioning temporal extensions of GNNs (Kazemi et al., 2020; Barros et al., 2021; Xue et al., 2022; Xie et al., 2020), or focus on specific topics, like temporal link prediction (Qin & Yeung, 2022; Skarding et al., 2021) or temporal graph generation (Gupta & Bedathur, 2022), or present an overview of GNN models designed for different types of graphs without providing in-depth coverage of temporal GNNs (Thomas et al., 2022). This work aims to fill this gap by providing a systematization of existing GNN-based methods for temporal graphs and a formalization of the tasks being addressed. Our main contributions are the following:

- We propose a coherent formalization of the different learning settings and of the tasks that can be performed on temporal graphs, unifying existing formalism and informal definitions that are scattered in the literature, and highlighting substantial gaps in what is currently being tackled;

- We organize existing TGNN works into a comprehensive taxonomy that groups methods according to the way in which time is represented and the mechanism with which it is taken into account;

- We highlight the limitations of current TGNN methods, discuss open challenges that deserve further investigation and present critical real-world applications where TGNNs could provide substantial gains.

# 2 Temporal Graphs

We provide a formal definition of the different types of graphs analyzed in this work and we structure different existing notions in a common framework.

**Definition 1 (Static Graph - SG)** *A* Static Graph *is a tuple $G = (V, E, X^V, X^E)$, where $V$ is the set of nodes, $E \subseteq V \times V$ is the set of edges, and $X^V, X^E$ are $d_V$-dimensional node features and $d_E$-dimensional edge features.*

Node and edge features may be empty. In the following, we assume that all graphs are directed, i.e., $(u, v) \in E$ does not imply that $(v, u) \in E$. Moreover, given $v \in V$, the set

$$\mathcal{N}[v] := \{u \in V : (u, v) \in V\},$$

denotes the neighborhood of $v$ in $G$.

Extending Qin & Yeung (2022), we define Temporal Graphs as follows.

**Definition 2 (Temporal Graph - TG)** *A* Temporal Graph *is a tuple $G_T = (V, E, V_T, E_T)$, where $V$ and $E$ are, respectively, the set of all possible nodes and edges appearing in a graph at any time, while*

$$V_T := \{(v, x^v, t_s, t_e) : v \in V, x^v \in \mathbb{R}^{d_V}, t_s \leq t_e\},$$
$$E_T := \{(e, x^e, t_s, t_e) : e \in E, x^e \in \mathbb{R}^{d_E}, t_s \leq t_e\},$$

*are the temporal nodes and edges, with time-dependent features and initial and final timestamps. A set of temporal graphs is denoted as $\mathcal{G}_{\mathcal{T}}$.*

Observe that we implicitly assume that the existence of a temporal edge in $E_T$ requires the simultaneous existence of the corresponding temporal nodes in $V_T$. Moreover, the definition implies that node and edge features are constant inside each interval $[t_s, t_e]$, but may otherwise change over time. Since the same node or edge may be listed multiple times, with different timestamps, we denote as $\bar{t}_s(v) = \min\{t_s : (v, x^v, t_s, t_e) \in V_T\}$ and $\bar{t}_e(v) = \max\{t_e : (v, x^v, t_s, t_e) \in V_T\}$ the time of first and last appearance of a node, and similarly for $\bar{t}_s(e), \bar{t}_e(e), e \in E$. Moreover, we set $T_s(G_T) := \min\{\bar{t}_s(v) : v \in V\}$, $T_e(G_T) := \max\{\bar{t}_e(v) : v \in V\}$ as the initial and final timestamps in a TG $G_T$. For two TGs $G_T^i := (V^i, E^i, V_T^i, E_T^i)$, $i = 1, 2$, we write $G_T^1 \subseteq_V G_T^2$ to indicate the topological inclusion $V^1 \subseteq V^2$, while no relation between the corresponding timestamps is required.

Given $v \in V$, the set

$$\mathcal{N}_t[v] := \{u \in V : \exists \ (e, x^e, t_s, t_e) \in E_T \text{ with } e = (u, v), t_s \leq t\}$$

is the temporal neighborhood of $v$ at time $t$. i.e., the list of nodes that have been connected to $v$ until time $t$.

General TGs have no restriction on their timestamps, which can take any value (for simplicity, we just assume that they are non-negative). However, in some applications, it makes sense to force these values to be multiples of a fixed time-step. This leads to the notion of Discrete Time Temporal Graphs, which are defined as follows.

**Definition 3 (Discrete Time Temporal Graph - DTTG)** *Let $\Delta t > 0$ be a fixed time-step and let $t_1 < t_2 < \cdots < t_n$ be timestamps with $t_{k+1} = t_k + \Delta t$. A* Discrete Time Temporal Graph $G_{DT}$ *is a TG where for each $(v, x^v, t_s, t_e) \in V_T$ or $(e, x^e, t_s, t_e) \in E_T$, the timestamps $t_s, t_e$ are taken from the set of fixed timestamps (i.e., $t_s, t_e \in \{t_1, t_2, \ldots, t_n\}$, with $t_s < t_e$).*

## 2.1 Representation of temporal graphs

In the existing literature, dynamic graphs are often divided into DTTG (as in Definition 3) and continuous-time temporal graphs (CTTG) (or time sequence graphs), which are defined e.g. in (Kazemi et al., 2020; Barros et al., 2021; Luo & Li, 2022; Gupta & Bedathur, 2022).

However, we find that this separation does not capture well the central difference between various graph characterizations, which is rather based on the fact that the data are represented as a stream of static graphs, or as a stream of single node and edge addition and deletion events. We thus formalize the following two categories for the description of time-varying graphs, based on snapshots or on events. These different representations lead to different algorithmic approaches and become particularly useful when organizing the methods in a taxonomy.

The snapshot-based strategy focuses on the temporal evolution of the whole graph. Snapshot-based Temporal Graphs can be defined as follows.

**Definition 4 (Snapshot-based Temporal Graph - STG)** *Let $t_1 < t_2 < \cdots < t_n$ be the ordered set of all timestamps $t_s, t_e$ occurring in a TG $G_T$. Set*

$$V_i := \{(v, x^v) : (v, x^v, t_s, t_e) \in V_T, t_s \leq t_i \leq t_e\},$$
$$E_i := \{(e, x^e) : (e, x^e, t_s, t_e) \in E_T, t_s \leq t_i \leq t_e\},$$

*and define the* snapshots $G_i := (V_i, E_i)$, $i = 1, \ldots, n$. *Then a* Snapshot-based Temporal Graph *representation of $G_T$ is the sequence*

$$G_T^S := \{(G_i, t_i) : i = 1, \ldots, n\}$$

*of time-stamped static graphs.*

This representation is mostly used to describe DTTGs, where the snapshots represent the TG captured at periodic intervals (e.g., hours, days, etc.).

The event-based strategy is instead more appropriate when the focus is on the temporal evolution of individual nodes or edges. This leads to the following definition.

**Definition 5 (Event-based Temporal Graph - ETG)** *Let $G_T$ be a TG, and let $\varepsilon$ denote one of the following* events*:*

- Node insertion $\varepsilon_V^+ := (v, t)$*: the node $v$ is added to $G_T$ at time $t$, i.e., there exists $(v, x^v, t_s, t_e) \in V_T$ with $t_s = t$.*

- Node deletion $\varepsilon_V^- := (v, t)$*: the node $v$ is removed from $G_T$ at time $t$, i.e., there exists $(v, x^v, t_s, t_e) \in V_T$ with $t_e = t$.*

- Edge insertion $\varepsilon_E^+ := (e, t)$*: the edge $e$ is added to $G_T$ at time $t$, i.e., there exists $(e, x^e, t_s, t_e) \in E_T$ with $t_s = t$.*

- Edge deletion $\varepsilon_E^- := (e, t)$*: the edge $e$ is removed from $G_T$ at time $t$, i.e., there exists $(e, x^e, t_s, t_e) \in E_T$ with $t_e = t$.*

*An* Event-based Temporal Graph *representation of TG is a sequence of events*

$$G_T^E := \{\varepsilon : \varepsilon \in \{\varepsilon_V^+, \varepsilon_V^-, \varepsilon_E^+, \varepsilon_E^-\}\}.$$

Here it is implicitly assumed that node and edge events are consistent (e.g., a node deletion event implies the existence of an edge deletion event for each incident edge). In the case of an ETG, the TG structure can be recovered by coupling an insertion and deletion event for each temporal edge and node. ETGs are better suited than STGs to represent TGs with arbitrary timestamps.

We will use the general notion of TG, which comprises both STG and ETG, in formalizing learning tasks in the next section. On the other hand, we will revert to the STG and ETG notions when introducing the taxonomy of TGNN methods in Section 6, since TGNNs use one or the other representation strategy in their algorithmic approaches.

## 3 Basic notions on Graph Neural Networks

GNNs are a class of neural network architectures specifically designed to process and analyze graph-structured data; they learn a function $\mathbf{h}_v = \text{GNN}(v, G; \theta)$, with $v \in V$ and $\theta$ being a set of trainable parameters. GNNs rely on the so called message passing mechanism, which implements a local computational scheme to process graphs. Formally, the information related to a node $v$ is stored into a feature vector $\mathbf{h}_v$ that is iteratively updated by aggregating the features of its neighboring nodes. After $k$ iterations, the vector $\mathbf{h}_v^k$ contains both the structural information and the node content of the $k$–hop neighborhood of $v$. Given a sufficient number of iterations, the node feature vectors can be used to classify the nodes or the entire graph. Specifically, the output of the $k$-th layer of a message passing GNN is:

$$\mathbf{h}_v^k = \text{COMBINE}^{(k)}(\mathbf{h}_v^{k-1}, \text{AGGREGATE}^{(k)}(\{\mathbf{h}_u^{k-1},\ u \in \mathcal{N}[v]\})) \tag{1}$$

where $\text{AGGREGATE}^{(k)}$ is a function that aggregates the node features from the neighborhood $\mathcal{N}[v]$ at the $(k-1)$-th iteration, and $\text{COMBINE}^{(k)}$ is a function that combines the features of the node $v$ with those of its

neighbors. The aggregation step often involves employing permutation invariant operations such as mean, max-pooling, and sum. These operations ensure that the final aggregated representation is insensitive to the ordering of the nodes. Instead, typical choices for the `COMBINE` function are concatenation and summation. In node level tasks, a `READOUT` function is used to produce an output for each node, based on its features at the final layer $K$:

$$\mathbf{o}_v \ = \ \mathtt{READOUT}(\mathbf{h}_v^K)$$

whereas, in graph level tasks, a `READOUT` function produces the final output given the feature vectors from the last layer $K$:

$$\mathbf{o} \ = \ \mathtt{READOUT}(\{\mathbf{h}_v^K, \ v \in V\}).$$

In our work we explore models that specifically tackle temporal graphs employing this kind of message passing scheme.

## 4    Other approaches to model temporal graphs

Traditionally, machine learning models for graphs have been mostly designed for static graphs (Xia et al., 2021; Zhang et al., 2020). However, many applications involve temporal graphs (Kumar et al., 2018; Dasgupta et al., 2018; Taheri et al., 2019). This introduces important challenges for learning and inference since nodes, attributes, and edges change over time. Many different representation techniques for temporal graphs have been recently proposed, apart from GNN-based solutions which leverage the message passing architecture (Section 3).

A popular class of approaches for learning an embedding function for temporal graphs is the class of random walk-based methods. For example, in Wang et al. (2021b) temporal random walks are exploited to efficiently and automatically sample temporal network motifs, i.e., connected subgraphs with links that appear within a restricted time range. Similarly, in Liu et al. (2020), a time-reinforced random walk is proposed to effectively sample the structural and temporal contexts over graph evolution. Also, Jin et al. (2022) employs spatiotemporal-biased random walks to identify a collection of representative motifs, enabling the effective characterization of temporal nodes. With DyANE (Sato et al., 2019), the temporal graphs are transformed into a static graph representation called a supra-adjacency representation. In this approach, the nodes are defined as (node, time) pairs from the original temporal graph. This static graph representation retains the temporal paths of the original network, which are crucial for comprehending and constraining the underlying dynamical processes. Afterwards, standard embedding techniques for static graphs, utilizing random walks, are employed.

Temporal graph learning has leveraged the use of temporal point processes as well. Temporal point processes are stochastic processes employed for modeling sequential asynchronous discrete events occurring in continuous time (Lewis, 1972). DyRep (Trivedi et al., 2019) is capable of learning a set of functions that can effectively generate evolving, low-dimensional node embeddings. By using the obtained node embeddings, a temporal point process is employed to estimate the likelihood of an edge connecting two nodes at a certain timestamp. In Trivedi et al. (2017), instead, the occurrence of an edge in a temporal graph is modeled as a multivariate point process, where the intensity function is influenced by the score assigned to that edge, which is computed using the learned entity embeddings. The entity embeddings, which evolve over time, are acquired through a recurrent architecture.

Non-negative matrix factorization (NMF) has been employed for the purpose of link prediction in temporal graphs. In Ahmed et al. (2018), novel iterative rules for NMF are proposed to construct the matrix factors that capture crucial features of the temporal graph, enhancing the accuracy of the link prediction process.

Moreover, statistical approaches have been applied to TGs; for example, Daniele et al. (2022) introduces a novel whiteness hypothesis test specifically designed for spatio-temporal graphs. The test extends traditional methods used for system identification within graph signals to detect dependencies among temporal observations and spatial dependencies among graph neighborhoods. The test can also be used to assess the optimality of forecasting models.

Lastly, the majority of recently proposed methods employ deep learning techniques. For example, DynGem (Goyal et al., 2018) is a dynamical autoencoder for growing graphs that construct the embedding of a snapshot based on the embedding of the previous snapshot. TRRN (Xu et al., 2021), instead, uses multi-head self-attention to process a set of memories, enabling efficient information flow from past observations to current latent representations through shortcut paths. It incorporates policy networks with differentiable binary routers to estimate the activation probability of each memory and dynamically update them at the most relevant time steps. In Xu et al. (2019), a spatio-temporal attentive recurrent network model, called STAR, is proposed for interpretable temporal node classification. In Opolka et al. (2019), a node-level regression task is achieved by training embeddings to maximize the mutual information between patches of the graph, at any given time step, and between features of the central nodes of patches, in the future. In Zhou et al. (2022a), a spectral-based solution for learning representations of long-range interactions is proposed. They utilize efficient spectral transforms and graph convolutions to capture temporal features and interactions. The approach addresses challenges in ETG learning and achieves well-conditioned embeddings with minimal information loss. TSNet (Zheng et al., 2021) is a comprehensive framework for node classification in temporal graphs, consisting of two key steps. Firstly, the graph snapshots undergo a sparsification process using edge sampling, guided by a learned distribution derived from the supervised classification task. This step effectively reduces the density of the snapshots. Subsequently, the sparsified snapshots are aggregated and processed through a convolutional network to extract meaningful features for node classification. Finally, Marisca et al. (2022) introduces a novel class of attention-based architectures called Spatiotemporal Point Inference Network (SPIN) for addressing the challenge of reconstructing multivariate time-series on sparse graphs with missing data. SPIN exploits a spatiotemporal propagation process to learn predictive representations of unobserved samples, taking into account the data missingness. By incorporating a hierarchical attention mechanism, the proposed method reduces the space and time complexities involved.

Instead of using graphs to represent the topological structure, time-dependent relational data may be modelled using different data structures, such as hypergraphs, heterogeneous networks, or multiplex networks (Battiston et al., 2020; Bianconi, 2021). Even if surveying this domain goes beyond the scope of this work, we mention that different recent solutions exist for learning on these structures (Agarwal et al., 2022; Behrouz et al., 2023; Fan et al., 2022; Wang et al., 2023).

In our survey, we aim to explore and analyze methods that leverage and adapt the GNN framework to temporal graphs. By delving into this specific subset of techniques, we seek to gain a deeper understanding of their applicability, effectiveness, and potential in capturing the temporal dynamics of complex graph structures.

# 5 Learning tasks on temporal graphs

Thanks to their learning capabilities, TGNNs are extremely flexible and can be adapted to a wide range of tasks on TGs. Some of these tasks are straightforward temporal extensions of their static counterparts. However, the temporal dimension has some non-trivial consequences in the definition of learning settings and tasks, some of which are often only loosely formalized in the literature. We start by formalizing the notions of transductive and inductive learning for TGNNs, and then describe the different tasks that can be addressed.

## 5.1 Learning settings

The machine learning literature distinguishes between inductive learning, in which a model is learned on training data and later applied to unseen test instances, and transductive learning, in which the input data of both training and test instances are assumed to be available, and learning is equivalent to leveraging the training inputs and labels to infer the labels of test instances given their inputs. This distinction becomes extremely relevant for graph-structured data, where the topological structure gives rise to a natural connection between nodes, and thus to a way to propagate the information in a transductive fashion. Roughly speaking, transductive learning is used in the graph learning literature when the node to be predicted and its neighborhood are known at training time — and is typical of node classification tasks —, while inductive

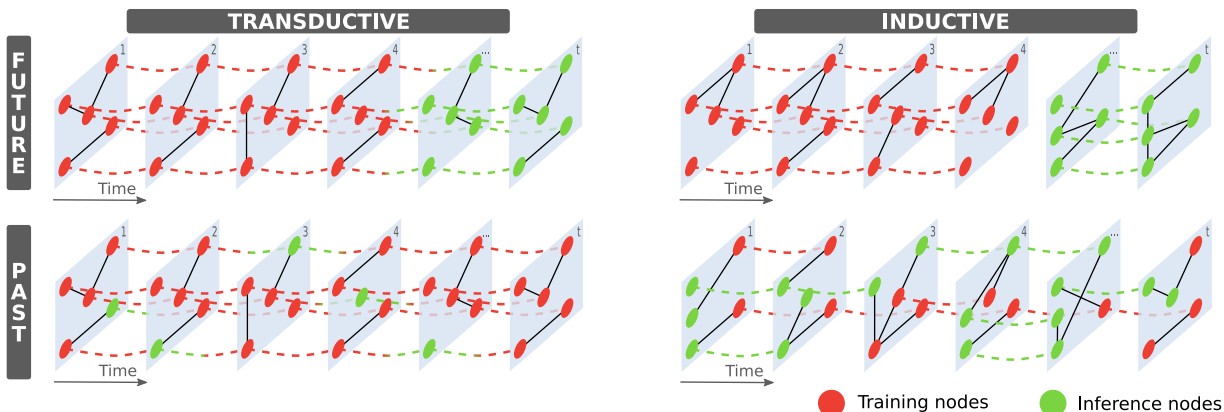

Figure 1: **Learning settings.** Schematic representation of the learning settings on TGs formalized in Section 5.1. The temporal graphs are represented as sequences of snapshots, with training (red) and inference (green) nodes connected by edges (solid lines), and where a dotted line connects instances of the same node (with possibly different features and/or labels) in successive snapshots. The four categories are obtained from the different combinations of a temporal and a topological dimension. The **temporal** dimension distinguishes the *future* setting, where the training nodes are all observed before the inference nodes (first row), from the *past* setting where inference is performed also on nodes appearing before the observation of the last training node (second row). The **topological** dimension comprises a *transductive* setting, where each inference node is observed (unlabelled) also during training (left column), and an *inductive* setting, where inference is performed on nodes that are unknown at training time (right column).

learning indicates that this information is not available — and is most often associated to graph classification tasks.

However, when talking about GNNs with their representation learning capabilities, this distinction is not so sharp. For example, a GNN trained for node classification in transductive mode could still be applied to an unseen graph, thus effectively performing inductive learning. The temporal dimension makes this classification even more elusive, since the graph structure is changing over time and nodes are naturally appearing and disappearing. Defining node membership in a temporal graph is thus a challenging task in itself.

Below, we provide a formal definition of transductive and inductive learning for TGNNs which is purely topological, i.e., linked to knowing or not the instance to be predicted at the training time, and we complete it with a temporal dimension, which distinguishes between past and future prediction tasks. A schematic representation of these settings is visualized in Figure 1. We recall (see Section 2) that $T_e(G_T)$ is the final timestamp in a TG $G_T$.

**Definition 6 (Learning settings)** *Assume that a model is trained on a set of $n \geq 1$ temporal graphs* $\mathcal{G}_{\mathcal{T}} := \{G_T^i := (V_i, E_i, X_i^V, X_i^E), \ i = 1, \dots, n\}$. *Moreover, let*

$$T_e^{all} := \max_{i=1,\dots,n} T_e(G_T^i), \ V^{all} := \cup_{i=1}^n V_i, \ E^{all} := \cup_{i=1}^n E_i,$$

*be the final timestamp and the set of all nodes and edges in the training set. Then, we have the following settings:*

- Transductive learning*: inference can only be performed on $v \in V^{all}$, $e \in E^{all}$, or $G_T \subseteq_V G_T^i$ with $G_T^i \in \mathcal{G}_{\mathcal{T}}$.*

- Inductive learning*: inference can be performed also on $v \notin V^{all}$, $e \notin E^{all}$, or $G_T \not\subseteq_V G_T^i$, for all $i = 1, \dots, n$.*

- Past prediction*: inference is performed for $t \leq T_e^{all}$.*

- Future prediction: *inference is performed for $t > T_e^{all}$.*

We remark that all combinations of topological and temporal settings are meaningful, except for the case of inductive graph-based tasks. Indeed, the measure of time used in TGs is relative to each single graph. Moving to an unobserved graph would thus make the distinction between past and future pointless. Moreover, let us observe that, in all other cases, the two temporal settings are defined based on the final time of the entire training set, and not of the specific instances (nodes or edges), since their embedding may change also as an effect of the change of their neighbors in the training set.

We will use this categorization to describe supervised and unsupervised learning tasks in Section 5.2-5.3, and to present existing models in Section 6.

## 5.2 Supervised learning tasks

Supervised learning tasks are based on a dataset where each object is annotated with its label (or class), from a finite set of possible choices $\mathcal{C} := \{C_1, C_2, \ldots, C_k\}$.

### 5.2.1 Classification

**Definition 7 (Temporal Node Classification)** *Given a TG $G_T = (V, E, V_T, E_T)$, the node classification task consists in learning the function*

$$f_{NC} : V \times \mathbb{R}^+ \to \mathcal{C}$$

*which maps each node to a class $C \in \mathcal{C}$, at a time $t \in \mathbb{R}^+$.*

This is one of the most common tasks in the TGNN literature. For instance, Pareja et al. (2020); Xu et al. (2020); Wang et al. (2021a); Zhou et al. (2022b); Rossi et al. (2020) focus on a future-transductive (FT) setting, i.e., predicting the label of a node in future timestamps. TGAT (Xu et al., 2020) performs future-inductive (FI) learning, i.e., it predicts the label of an unseen node in the future. Finally, DGNN (Ma et al., 2020) is the only method that has been tested on a past-inductive (PI) setting, i.e., predicting labels of past nodes that are unavailable (or masked) during training, while no approach has been applied to the past-transductive (PT) one. A significant application may be in epidemic surveillance, where contact tracing is used to produce a TG of past human interactions, and sample testing reveals the labels (infection status) of a set of individuals. Identifying the past infection status of the untested nodes is a PT task.

**Definition 8 (Temporal Edge Classification)** *Given a TG $G_T = (V, E, V_T, E_T)$, the temporal edge classification task consists in learning a function*

$$f_{EC} : E \times \mathbb{R}^+ \to \mathcal{C}$$

*which assigns each edge to a class at a given time $t \in \mathbb{R}^+$.*

Temporal edge classification has been less explored in the literature. Existing methods have focused on FT learning (Pareja et al., 2020; Wang et al., 2021a), while FI, PI and PT have not been tackled so far. An example of PT learning consists in predicting the unknown past relationship between two acquaintances in a social network given their subsequent behavior. For FI, one may predict if a future transaction between new users is a fraud or not.

In the next definition we use the set of real and positive intervals $I^+ := \{[t_s, t_e] \subset \mathbb{R}^+\}$.

**Definition 9 (Temporal Graph Classification)** *Let $\mathcal{G}_\mathcal{T}$ be a domain of TGs. The graph classification task requires to learn a function*

$$f_{GC} : \mathcal{G}_\mathcal{T} \times I^+ \to \mathcal{C}$$

*that maps a temporal graph, restricted to a time interval $[t_s, t_e] \in I^+$, into a class.*

The definition includes the classification of a single snapshot (i.e., $t_s = t_e$). As mentioned above, in the inductive setting the distinction between past and future predictions is pointless. In the transductive setting,

instead, a graph $G_T \in \mathcal{G}_\mathcal{T}$ may be classified in a past mode if $[T_s(G_T), T_e(G_T)] \subseteq [t_s, t_e]$, or in the future mode, otherwise.

The only existing method addressing the classification of temporal graphs is found in Micheli & Tortorella (2022), where the discrimination between STGs characterized by different dissemination processes is formalized as a PT classification task. The temporal graph classification task can have numerous relevant applications. For instance, an example of inductive temporal graph classification is predicting mental disorders from the analysis of the brain connectome (Heuvel et al., 2010). On the other hand, detecting critical stages during disease progression from gene expression profiles (Gao et al., 2022c) can be framed as a past transductive graph classification task.

### 5.2.2 Regression

The tasks introduced for classification can all be turned into corresponding regression tasks, simply by replacing the categorical target $\mathcal{C}$ with the set $\mathbb{R}$. We omit the formal definitions for the sake of brevity. Static GNNs have already shown outstanding results in this setting, e.g., in weather forecasting (Keisler, 2022) and earthquake location and estimation (McBrearty & Beroza, 2022). However, limited research has been conducted on the application of TGNNs to regression tasks. Notable exceptions are the use of TGNNs in two FT regression tasks, the traffic prediction (Cini et al., 2022) and the prediction of the incidence of chicken pox cases in neighboring countries (Micheli & Tortorella, 2022).

### 5.2.3 Link prediction

Link prediction requires the model to predict the relation between two given nodes, and can be formulated by taking as input any possible pair of nodes. Thus, we consider the setting to be transductive when both node instances are known at training time, and inductive otherwise. Instead, Qin & Yeung (2022) adopt a different approach and identify *Level-1* (the set of nodes is fixed) and *Level-2* (nodes may be added and removed over time) temporal link prediction tasks.

**Definition 10 (Temporal Link Prediction)** *Let $G_T = (V, E, V_T, E_T)$ be a TG. The temporal link prediction task consists in learning a function*

$$f_{LP} : V \times V \times \mathbb{R}^+ \to [0, 1]$$

*which predicts the probability that, at a certain time, there exists an edge between two given nodes.*

The domain of the function $f_{\text{LP}}$ is the set of all feasible pairs of nodes, since it is possible to predict the probability of future interactions between nodes that have been connected in the past or not, as well as the probability of missing edges in a past time. Most TGNN approaches for temporal link prediction focus on future predictions, forecasting the existence of an edge in a future timestamp between existing nodes (FT is the most common setting) (Pareja et al., 2020; Sankar et al., 2020; Hajiramezanali et al., 2019; You et al., 2022; Xu et al., 2020; Luo & Li, 2022; Wang et al., 2021a; Ma et al., 2020; Rossi et al., 2020; Zhou et al., 2022b), or unseen nodes (FI) (Hajiramezanali et al., 2019; Xu et al., 2020; Rossi et al., 2020). The only model that investigates past temporal link prediction is Luo & Li (2022), which devises a PI setting by masking some nodes and predicting the existence of a past edge between them. Note that predicting past temporal links can be extremely useful for predicting, e.g., missing interactions in contact tracing for epidemiological studies.

**Definition 11 (Event Time Prediction)** *Let $G_T = (V, E, V_T, E_T)$ be a TG. The aim of the event time prediction task is to learn a function*

$$f_{EP} : V \times V \to \mathbb{R}^+$$

*that predicts the time of the first appearance of an edge.*

None of the existing methods address this task. Potential FT applications of event time prediction include predicting when a customer will pay an invoice to its supplier, or how long it takes to connect two similar users in a social network.

### 5.3 Unsupervised learning tasks

In this section, we formalize unsupervised learning tasks on temporal graphs, an area that has received little to no attention in the TGNN literature so far.

#### 5.3.1 Clustering

Temporal graphs can be clustered at the node or graph level, with edge-level clustering being a minor variation of the node-level one. Some relevant applications can be defined in terms of temporal clustering.

**Definition 12 (Temporal Node Clustering)** *Given a TG $G_T = (V, E, V_T, E_T)$, the temporal node clustering task consists in learning a time-dependent cluster assignment map*

$$f_{NCl} : V \times \mathbb{R}^+ \to \mathcal{P}(V),$$

*where $\mathcal{P}(V) := \{p_1, p_2, \ldots, p_k\}$ is a partition of the node set $V$, i.e., $p_i \subset V_T$, $p_i \cap p_j = \emptyset$, if $i \neq j$, $\cup_{i=1}^N p_i = V_T$.*

While node clustering in SGs is a very common task, its temporal counterpart has not been explored yet for TGNNs, despite its potential relevance in application domains like epidemic modelling, e.g., identifying groups of exposed individuals, in both inductive and transductive settings (Ru et al., 2023; Hiram Guzzi et al., 2022; Koher et al., 2019; Darbon et al., 2019; 2018), trend detection in customer profiling, mostly in transductive settings (Ljubičić et al., 2022; Rosyidah et al., 2019), or disease clustering, mostly in future transductive settings (Jacquez et al., 2019; Meliker et al., 2009; Wheeler, 2007; Ozonoff et al., 2005).

**Definition 13 (Temporal Graph Clustering)** *Given a set of temporal graphs $\mathcal{G}_\mathcal{T}$, the temporal graph clustering task consists in learning a cluster-assignment function*

$$f_{GCl} : \mathcal{G}_\mathcal{T} \times I^+ \to \mathcal{P}(\mathcal{G}_\mathcal{T}),$$

*where $\mathcal{P}(\mathcal{G}_\mathcal{T}) := \{p_1, \ldots, p_k\}$ is a partition of the set of temporal graphs in the given time interval.*

Relevant examples of tasks of inductive temporal graph clustering are grouping social interaction networks (e.g., hospitals, workplaces, schools) according to their interaction patterns (Longa et al., 2022c), or grouping diseases in terms of similarity between their spreading processes (Enright & Rowland, 2018; Myers et al., 2023).

#### 5.3.2 Anomaly detection

Anomaly detection in TGs refers to the process of identifying any significant deviations from the expected or normal patterns of connectivity, behavior, or structural properties within the TG. These anomalies may indicate critical events, emerging trends, unusual behaviors, or potential threats. The basic idea is to model the behavior of the data using density estimation methods and then identify instances that deviate significantly from this behavior as anomalies. In the following we identify and formalize three types of anomalies.

**Definition 14 (Anomalous node detection)** *Given a TG $G_T = (V, E, V_T, E_T)$, a TGNN and an application dependent threshold $C \in (0, 1)$, a node $v \in V$ is detected as anomalous if*

$$f_{DE}(\texttt{TGNN}(v, G_T; \theta)) < C,$$

*where $f_{DE}$ is a procedure for density estimation like variational autoencoder (An & Cho, 2015), generative adversarial network (Schlegl et al., 2017) or kernel density estimation (Kim & Scott, 2012).*

Fo example, detection of anomaly nodes within a TG can be applied to identify the origin nodes of a virus in a network. By comparing the activity of each node on the days surrounding a known virus attack to their

activity on the day of the attack, the nodes that exhibit higher-than-usual activity at the time of the attack can be detected as the potential source of the virus (Ranshous et al., 2015). Other applications include discovering anomalies in communication networks (Akoglu & Faloutsos, 2010) and observing the shifts in community involvement (Khan & Haroon, 2022; Gao et al., 2010; Ji et al., 2013).

**Definition 15 (Anomalous edge detection)** *Given a TG $G_T = (V, E, V_T, E_T)$, a TGNN, an application dependent threshold $C \in (0, 1)$ and a procedure for density estimation $f_{DE}$, an edge $e \in E$ is detected as anomalous if*

$$f_{DE}(\textit{TGNN}(e, G_T; \theta)) < C.$$

Edge anomaly detection finds extensive applications in various fields, such as the analysis of vehicle traffic patterns (Zhang et al., 2018; Deng et al., 2022) and the identification of improbable social interactions (Savage et al., 2014; Heard et al., 2010).

**Definition 16 (Anomalous graph detection)** *Given a domain of TGs $\mathcal{G}_{\mathcal{T}}$, a TGNN, an application dependent threshold $C \in (0, 1)$ and a procedure for density estimation $f_{DE}$, a TG $G_T \in \mathcal{G}_{\mathcal{T}}$ is detected as anomalous if*

$$f_{DE}(\textit{TGNN}(G_T; \theta)) < C.$$

Possible application of anomalous graph detection are identifying accidents in TGs representing vehicle traffic (Ma et al., 2021; Mongiovi et al., 2013) as well as detecting whether a molecule is mutagenic (Zambon et al., 2018).

### 5.3.3 Low-dimensional embedding (LDE)

LDEs are especially useful in the temporal setting, e.g., to visually inspect temporal dynamics of individual nodes or entire graphs, and identify relevant trends and patterns. No GNN-based models have been applied to these tasks, neither at the node nor at the graph level. We formally define the tasks of temporal node and graph LDE as follows.

**Definition 17 (Low-dimensional temporal node embedding)** *Given a TG $G_T = (V, E, V_T, E_T)$, the low-dimensional temporal node embedding task consists in learning a map*

$$f_{NEm} : V \times \mathbb{R}^+ \to \mathbb{R}^d$$

*to map a node, at a given time, into a low dimensional space.*

**Definition 18 (Low-dimensional temporal graph embedding)** *Given a domain of TGs $\mathcal{G}_{\mathcal{T}}$, the low-dimensional temporal graph embedding task aims to learn a map*

$$f_{GEm} : \mathcal{G}_{\mathcal{T}} \times I^+ \to \mathbb{R}^d,$$

*which represents each graph as a low dimensional vector in a given time interval.*

## 6 A taxonomy of TGNNs

This section describes the taxonomy with which we categorize existing TGNN approaches (see Figure 2). All these methods learn a time-dependent embedding $\mathbf{h}_v(t) = \textit{TGNN}(v, G_T; \theta)$ of each node $v \in V_T$ of a TG $G_T$, where again $\theta$ represents a set of trainable parameters. Following the representation strategies outlined in Section 2.1, the first level groups methods into *Snapshot-based* and *Event-based*. The second level of the taxonomy further divides these two macro-categories based on the techniques used to manage the temporal

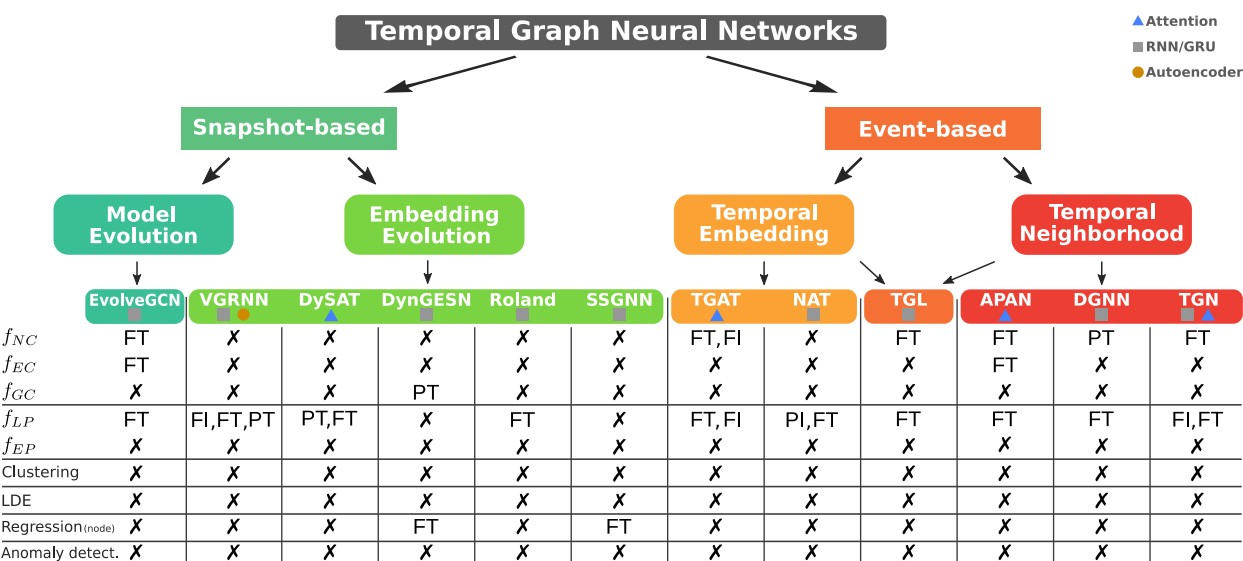

Figure 2: **The proposed TGNN taxonomy and an analysis of the surveyed methods.** The top panel shows the new categories introduced in this work with the corresponding model instances (Section 6), where the colored bullets additionally indicate the main technology that they employ. The bottom table maps these methods to the task (Section 5) to which they have been applied in the respective original paper, with an additional indication of their use in the future (F), past (P), inductive (I), or transductive (T) settings (Section 5.1). Notice that no method has been applied yet to clustering and visualization, for neither graphs nor nodes. Moreover, only four out of ten models have been tested in the past mode (three in PT, one in PI).

dependencies. The leaves of the taxonomy in Figure 2 correspond to the individual models, with a colored symbol indicating their main underlying technology.

In the following we will denote as $\mathtt{REC}(v_1, \ldots, v_k)$ a network that can process streams of tensors $v_1, \ldots, v_k$ and predict the next one in the sequence. This is usually a Recurrent Neural Network (from which we set the name $\mathtt{REC}$), but other mechanisms such as temporal attention can be used.

## 6.1 Snapshot-based models

*Snapshot-based* models are specifically tailored for STGs (see Def. 4) and thus, consistently with the definition, they are equipped with a suitable method to process the entire graph at each point in time, and with a mechanism that learns the temporal dependencies across time-steps. Based on the mechanism used, we can further distinguish between *Model Evolution* and *Embedding Evolution* methods.

### 6.1.1 Model Evolution methods

We call *Model Evolution* the evolution of the parameters of a static GNN model over time. This mechanism is appropriate for modelling STG, as the evolution of the model is performed at the snapshot level.

More formally, these methods learn an embedding $\mathbf{h}_v(t_i) = \mathtt{GNN}(v, G_i; \theta(t_i))$, where $\theta(t_i) = \mathtt{REC}(\theta(t_{i-j}) : 1 \leq j \leq i_{\max})$ is a parameter-evolution network, and $i_{\max}$ is the memory length.

To the best of our knowledge, the only existing method belonging to this category is **EvolveGCN** (Pareja et al., 2020). This model utilizes a Recurrent Neural Network (RNN) to update the Graph Convolutional Network (GCN) (Kipf & Welling, 2016a) parameters at each time-step, allowing for model adaptation that is not constrained by the presence or absence of nodes. The method can effectively handle new nodes without prior historical information. A key advantage of this approach is that the GCN parameters are no longer trained directly, but rather they are computed from the trained RNN, resulting in a more manageable

model size that does not increase with the number of time-steps. The paper presents two versions of this method: EvolveGCN-O uses a Long Short-Term Memory (LSTM) to simply evolve the weights in time, while EvolveGCN-H represents the weights as hidden states of a Gated Recurrent Unit (GRU), whose input is the previous node embedding.

### 6.1.2 Embedding Evolution methods

Rather than evolving the parameters of a static GNN model, *Embedding Evolution* methods focus on evolving the embeddings produced by a static model. This means to learn a node embedding $\mathbf{h}_v(t_i) = \text{REC}(\mathbf{h}_v(t_{i-j}), i = 1, \dots, t_{\max})$ as the evolution of previous embeddings, where $\mathbf{h}_v(t_{i-j}) = \text{GNN}(v, G_{i-j}; \theta)$ are GNN embeddings for the SG $G_{i-j}$.

There are several different TGNN models that fall under this category. These networks differ from one another in the techniques used for processing both the structural information and the temporal dynamics of the STGs. **DySAT** (Sankar et al., 2020) introduces a generalization of Graph Attention Network (GAT) (Veličković et al., 2017) for STGs. First, it uses a self-attention mechanism to generate static node embeddings at each timestamp. Then, it uses a second self-attention block to process past temporal embeddings for a node to generate its novel embedding. Decoupling graph evolution into two modular blocks allows for efficient computations of temporal node representations. The structural and temporal self-attention layers, combined and stacked, enable flexibility and scalability. The **VGRNN** model (Hajiramezanali et al., 2019) uses VGAE (Kipf & Welling, 2016b) on each snapshot, where the latent representations are conditioned on a state variable modelled by Semi-Implicit Variational Inference (SIVI) (Yin & Zhou, 2018) to handle the variation of the graph over time. The learned latent representation is then evolved through an LSTM conditioned on the previous time's latent representation, allowing the model to predict the future evolution of the graph.

**ROLAND** (You et al., 2022) is a general framework for extending state-of-the-art GNN techniques to STGs. The key insight is that node embeddings at different GNN layers can be viewed as hierarchical node states. To generalize a static GNN for dynamic settings, hierarchical node states are updated based on newly observed nodes and edges through a Gated Recurrent Unit (GRU) update module (Chung et al., 2014). The paper presents two versions of the model: ROLAND-MLP, which uses a 2-layer MLP to update node embeddings, and ROLAND-*moving average*, which updates the node embeddings through the moving average among previous node embeddings. Finally, reservoir computing techniques have also been proposed. **DynGESN** (Micheli & Tortorella, 2022) presents a method where each node embedding is updated by a recurrent mechanism using its temporal neighborhood and previous embedding, with fixed and randomly initialized recurrent weights. **SSGNN** (Cini et al., 2022) follows a similar approach but introduces trainable parameters in the decoder and combines randomized components in the encoder: initially, the encoder creates representations of the time series data observed at each node, by utilizing a reservoir that captures dynamics at various time scales; these representations are then further processed to incorporate spatial dynamics dictated by the graph structure.

## 6.2 Event-based models

Models belonging to the *Event-based* macro category are designed to process ETGs (see Def. 5). These models are able to process streams of events by incorporating techniques that update the representation of a node whenever an event involving that node occurs, and they are an extension of message passing to TGs, since they combine and aggregate node representations over temporal neighborhoods.

The models that lie in this macro category can be further classified in *Temporal Embedding* and *Temporal Neighborhood* methods, based on the technology used to learn the time dependencies. In particular, the *Temporal Embedding* models use recurrent or self-attention mechanisms to model sequential information from streams of events, while also incorporating a time encoding. This allows for temporal signals to be modeled by the interaction between time embedding, node features and the topology of the graph. *Temporal Neighborhood* models, instead, use a module that stores functions of events involving a specific node at a given time. These values are then aggregated and used to update the node representation as time progresses.

### 6.2.1 Temporal Embedding methods

*Temporal embedding* methods model TGs by combining time embedding, node features, and graph topology. These models use an explicit functional time encoding, i.e., a vector embedding $\mathbf{g}_t$ of time based on Random Fourier Features (RFF) (Rahimi & Recht, 2008), which is translation-invariant (i.e., it depends only on the elapsed and not the absolute time).

They extend the message passing architecture to temporal neighborhoods, where the time is encoded by $\mathbf{g}_t$, i.e.,

$$\mathbf{h}_v(t) = \texttt{COMBINE}((\mathbf{h}_v(t), \mathbf{g}_0), \texttt{AGGREGATE}(\{(\mathbf{h}_u(t'), \mathbf{g}_{t-t'}), \, u \in \mathcal{N}_\mathcal{T}[v]\}))$$

where $t'$ is the time of the connection event between $u$ and $v$. Thus, $\mathbf{g}_{t-t'}$ encodes the time elapsed between the current time $t$ and the time of connection between $u$ and $v$.

**TGAT** (Xu et al., 2020), for example, introduces a graph-temporal attention mechanism which works on the embeddings of the temporal neighbors of a node, where the positional encoding is replaced by a temporal encoding based on RFFs. In addition, Xu et al. (2020) implement a version of TGAT with all temporal attention weights set to an equal value (Const-TGAT). On the other hand, **NAT** (Luo & Li, 2022) collects the temporal neighbors of each node into dictionaries, and then it learns the node representation with a recurrent mechanism, using the historical neighborhood of the current node and a RFF based time embedding. Note that Luo & Li (2022) propose a dedicated data structure to support parallel access and update of the dictionary on GPUs.

### 6.2.2 Temporal Neighborhood methods

The *Temporal Neighborhood* class includes all TGNN models that make use of a special *mailbox* module to update node embeddings based on events. When an event $\varepsilon$ occurs, a function is evaluated on the details of the event to compute a *mail* or a *message* $\mathbf{m}_\varepsilon$. For example, when a new edge appears between two nodes, a message is produced, taking into account the time of occurrence of the event, the node features, and the features of the new edge. The node representation is then updated at each time by aggregating all the generated messages. In more details, these methods extend message passing by learning an embedding

$$\mathbf{h}_v(t) = \texttt{COMBINE}(\mathbf{h}_v(t), \texttt{AGGREGATE}(\{\mathbf{m}_\varepsilon, \, \varepsilon = (u, t') \text{ with } u \in \mathcal{N}_\mathcal{T}[v]\})),$$

where $\varepsilon$, with $u \in \mathcal{N}_\mathcal{T}[v]$, is the addition or deletion of a temporal neighbor of $v$.

Several existing TGNN methods belong to this category. **APAN** (Wang et al., 2021a) introduces the concept of asynchronous algorithm, which decouples graph query and model inference. An attention-based encoder maps the content of the mailbox to a latent representation of each node, which is decoded by an MLP adapted to the downstream task. After each node update following an event, mails containing the current node embedding are sent to the mailboxes of its neighbors using a propagator. **DGNN** (Ma et al., 2020) combines an *interact* module — which generates an encoding of each event based on the current embedding of the interacting nodes and its history of past interactions — and a *propagate* module — which transmits the updated encoding to each neighbors of the interacting nodes. The aggregation of the current node encoding with those of its temporal neighbors uses a modified LSTM, which permits to work on non-constant time-steps, and implements a discount factor to downweight the importance of remote interactions. **TGN** (Rossi et al., 2020) provides a generic framework for representation learning in ETGs, and it makes an effort to integrate the concepts put forward in earlier techniques. This inductive framework is made up of separate and interchangeable modules. Each node the model has seen so far is characterized by a memory vector, which is a compressed representation of all its past interactions. Given a new event, a mailbox module computes a mail for every node involved. Mails will then be used to update the memory vector. To overcome the so-called staleness problem (Kazemi et al., 2020), an embedding module computes, at each timestamp, the node embeddings using their neighborhood and their memory states. Finally, **TGL** (Zhou et al., 2022b) is a general framework for training TGNNs on graphs with billions of nodes and edges by using a distributed training approach. In TGL, a mailbox module is used to store a limited number of the most recent interactions, called mails. When a new event occurs, the node memory of the relevant nodes is updated using the cached messages in the mailbox. The mailbox is then updated after the node embeddings

are calculated. This process is also used during inference to ensure consistency in the node memory, even though updating the memory is not required during this phase.

### 6.3 Category comparison

The categories of models identified in our taxonomy exhibit various strengths, weaknesses, or suitability for specific scenarios. First and foremost, the comparison between the macro categories of Snapshot-based and Event-based methods is straightforward, hinging on the choice of temporal graph representation, namely STGs or ETGs. Within each macro category, sub-categories exhibit their own set of advantages and disadvantages. For instance, in the Model Evolution category, learning the evolution of GNN parameters becomes complex when the GNN has a large number of parameters. On the other hand, the Embedding Evolution category has a limitation in that temporal learning exclusively relies on recurrent mechanisms, which may not fully guarantee the preservation of temporal correlations among substructures. Despite this drawback, the approach offers simplicity and intuitiveness, allowing for the exploration and evaluation of various recurrent mechanisms.

In Temporal Embedding methods, defining the time encoding function is not trivial because it should capture different aspects of the graph, such as the temporal periodicity of interactions, recurrent interactions over time, and more. One advantage of this category, though, is that ad-hoc time encoding functions can be defined, depending on the application domain.

Finally, for the Temporal Neighborhood category, constructing the mailbox can be complex. For example, dense graphs have large mailboxes, which necessitate managing scalability issues. A specific mechanism to decide which nodes to include in the mailbox needs to be carefully designed, and this mechanism may also be domain-dependent. Similarly to the Temporal Embedding category, a benefit of the Temporal Neighborhood models is their ability to define domain-specific mailbox mechanisms.

In summary, each category of models for temporal graph learning has its own set of advantages and disadvantages. Choosing a category depends on the representation of the input graph, the complexity of learning the temporal dynamics, the need for domain-specific encoding or mailbox mechanisms, and scalability considerations.

## 7 Open challenges

Building on existing libraries of GNN methods, two major TGNN libraries have been developed, namely PyTorch Geometric Temporal (PyGT) (Rozemberczki et al., 2021), based on PyTorch Geometric[1], and DynaGraph (Guan et al., 2022), based on Deep Graph Library[2]. While these are substantial contributions to the development and practical application of TGNN models, several open challenges still need to be faced to fully exploit the potential of this technology. We discuss the ones we believe are the most relevant in the following.

**Evaluation** The evaluation of GNN models has been greatly enhanced by the Open Graph Benchmark (OGB) (Hu et al., 2020), which provides a standardized evaluation protocol and a collection of graph datasets enabling a fair and consistent comparison between GNN models. An equally well-founded standardized benchmark for evaluating TGNNs does not currently exist, even if a promising first step in this direction is the recently published Temporal Graph Benchmark (TGB)[3]. As a result, each model has been tested on its own selection of datasets, making it challenging to compare and rank different TGNNs on a fair basis. For instance, Zhou et al. (2022b) introduced two real-world datasets with 0.2 billion and 1.3 billion temporal edges which allow to evaluate the scalability of TGNNs to large scale real-world scenarios, but they only tested the TGL model (Zhou et al., 2022b). The variety and the complexity of learning settings and tasks described in Section 5 makes a standardization of

---

[1]https://pytorch-geometric.readthedocs.io
[2]https://docs.dgl.ai/
[3]https://tgb.complexdatalab.com/

tasks, datasets and processing pipelines especially crucial to allow a fair assessment of the different approaches and foster innovation in the field.

Another crucial aspect of evaluating GNN models is explainability, which is the ability to interpret and understand their decision process. While explainability has been largely explored for standard GNNs (Luo et al., 2020; Ying et al., 2019; Longa et al., 2022a; Azzolin et al., 2022), only few works focused on explaining TGNNs (Xia et al., 2022; Vu & Thai, 2022; He et al., 2022).

**Expressiveness**  Driven by the popularity of (static) GNNs, the study of their expressive power has received a lot of attention in the last few years (Sato, 2020). For instance, appropriate formulations of message-passing GNNs have been shown to be as powerful as the Weisfeiler-Lehman isomorphism test (WL test) in distinguishing graphs or nodes (Xu et al., 2018), and higher-order generalizations of message-passing GNNs have been proposed that can match the expressivity of the $k$-WL test (Morris et al., 2019). Moreover, it has been proven that GNNs are a sort of universal approximators on graphs modulo the node equivalence induced by the WL test (D'Inverno et al., 2021). Finally, also the expressive power of GNNs equipped with pooling operators have been studied in Bianchi & Lachi (2023).

Conversely, the expressive power of TGNNs is still far from being fully explored, and the design of new WL tests, suitable for TGNNs, is a crucial step towards this aim. This is a challenging task since the definition of a node neighborhood in temporal graphs is not as trivial as for static graphs, due to the appearing/disappearing of nodes and edges. In Beddar-Wiesing et al. (2022), a new version of the WL test for temporal graphs has been proposed, applicable only to DTTGs. Instead, Souza et al. (2022) proposed a novel WL test for ETGs, and the TGN model (Rossi et al., 2020) has been proved to be as powerful as this test. Finally, Beddar-Wiesing et al. (2022) proved a universal approximation theorem, but the result just holds for a specific TGNN model for STGs, composed of standard GNNs stacked with an RNN.

To the best of our knowledge, these are the only results achieved so far on the expressive power of TGNNs. A complete theory of the WL test for the different TG representations, such as universal approximation theorems for event-based models, is still lacking. Moreover, no efforts have been made to incorporate higher-order graph structures to enhance the expressiveness of TGNNs. This task is particularly demanding, since it requires not only the definition of the temporal counterpart of the $k$-WL test but also some techniques to scale to large datasets. Indeed, a drawback of considering higher-order structures is that of high memory consumption, which can only get worse in the case of TGs, as they usually have a greater number of nodes than static graphs.

**Learnability**  Training standard GNNs over large and complex graph data is highly non-trivial, often resulting in problems such as over-smoothing and over-squashing. A theoretical explanation for this difficulty has been given using algebraic topology and Sheaf theory (Bodnar et al., 2022; Topping et al., 2021). More intuitively, we yet do not know how to reproduce the breakthrough obtained in training very deep architectures over vector data when training deep GNNs. Such a difficulty is even more challenging with TGNNs, because the typical long-term dependency of TGs poses additional problems to those due to over-smoothing and over-squashing.

Modern static GNN models face the problems arising from the complexity of the data using techniques such as dropout, virtual nodes, neighbor sampling, but a general solution is far from being reached. The extension of the above mentioned techniques to TGNNs, and the corresponding theoretical studies, are open challenges and we are aware of only one work towards this goal (Yang et al., 2020). On the other hand, the goal of proposing general very deep TGNNs is even more challenging due to the difficulty in designing the graph dynamics in a hierarchical fashion.

**Real-world applications**  The analysis of the tasks in Section 5 revealed several opportunities for the use of TGNNs far beyond their current scope of application. We would like to outline here some promising directions of application.

A challenging and potentially disruptive direction for the application of TGNNs is the learning of dynamical systems through the combination of machine learning and physical knowledge (Willard

et al., 2022). Physic Informed Neural Networks (PINNs) (Raissi et al., 2017) are already revolution-izing the field of scientific computing (Cuomo et al., 2022), and static GNNs have been employed in this framework with great success (Pfaff et al., 2021; Gao et al., 2022b). Adapting TGNNs to this field may enable to carry over these results to the treatment of time-dependent problems. Climate science (Faghmous & Kumar, 2014) is a particularly attractive field of application, both for its crit-ical impact in our societies and for the promising results achieved by GNNs in climate modelling tasks (Keisler, 2022). We believe that TGNNs may rise to be a prominent technology in this field, thanks to their unique capability to capture spatio-temporal correlations at multiple scales. Epi-demics studies are another topic of enormous everyday impact that may be explored through the lens of TGNNs, since a proper modelling of the spreading dynamics needs to be tightly coupled to the underlying TG structure (Enright & Rowland, 2018). Both fields requires a better development of TGNNs for regression problems, a task that is still underdeveloped (see Section 5).

# 8 Conclusion

GNN based models for temporal graphs have become a promising research area. However, we believe that the potential of GNNs in this field has only been partially explored. In this work, we propose a systematic formalization of tasks and learning settings for TGNNs, which was lacking in the literature, and a comprehensive taxonomy categorizing existing methods and highlighting unaddressed tasks. Building on this systematization of the current state-of-the-art, we discuss open challenges that need to be addressed to unleash the full potential of TGNNs. We conclude by stressing the fact that the issues open to date are very challenging, since they presuppose considering both the temporal and relational dimension of data, suggesting that forthcoming new computational models must go beyond the GNN framework to provide substantially better solutions.

# Acknowledgments

This research was partially supported by TAILOR, a project funded by EU Horizon 2020 research and innovation programme under GA No 952215. M.B. and V.L. are partially supported by the MUR PNRR project "THE - Tuscany Health Ecosystem" (spoke 3). P.L. acknowledges funding from the EU projects CHARM and TROPHY. B.L. and A.P. acknowledge the support of the PNRR project FAIR - Future AI Research (PE00000013), under the NRRP MUR program funded by the NextGenerationEU.

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
