# OpenReview forum: "Graph Neural Networks for Temporal Graphs: State of the Art, Open Challenges, and Opportunities"
_TMLR — Accepted by TMLR_

### Review · Reviewer_bWTz · 2023-06-02

**Summary Of Contributions:**

The paper provides an overview of Graph Neural Networks (GNNs) and their application to temporal graphs. It discusses the challenges associated with developing GNN-based models for temporal graphs, such as over-smoothing and over-squashing. The paper also reviews existing approaches to temporal GNNs and how they differ in terms of how the temporal aspect is represented and processed.

The paper is well-structured, with a clear introduction that sets out the scope and objectives of the paper. The authors provide a comprehensive literature review that covers recent research in the field of temporal GNNs. They also discuss open challenges in this area and identify potential opportunities for future research.

Overall, this paper provides a valuable resource for researchers interested in GNN-based models for temporal graphs. It offers insights into current research trends, identifies key challenges, and highlights potential areas for future exploration.



**Audience:**

Yes

**Broader Impact Concerns:**

No concerns on it.

**Claims And Evidence:**

Yes

**Requested Changes:**

As a survey, it is supposed to be comprehensive, so more related works are requested to be included for a thorough discussion.

===>comments after revision =====
concerns are well addressed.

**Strengths And Weaknesses:**


Pros:

The paper provides a comprehensive overview of Graph Neural Networks (GNNs) and their application to temporal graphs.
The authors provide a rigorous formalization of learning settings and tasks, which helps to clarify the scope of the paper.
The paper reviews existing approaches to temporal GNNs and categorizes them based on how they represent and process the temporal aspect.
The authors identify key challenges in developing GNN-based models for temporal graphs, such as over-smoothing and over-squashing.
The paper highlights potential opportunities for future research in this area.


Cons:
While the paper provides a comprehensive overview of existing approaches to temporal GNNs, it does not offer a detailed comparison or evaluation of these approaches.
Some clearly related papers are still missing, such as：
Transformer-Style Relational Reasoning with Dynamic Memory Updating for Temporal Network Modeling. The 2021 AAAI International Conference on Artificial Intelligence (AAAI’21), 2021
Node Classification in Temporal Graphs through Stochastic Sparsification and Temporal Structural Convolution. European Conference on Machine Learning and Principles and Practice of Knowledge Discovery in Databases (ECML-PKDD'20), 2020
Spatio-Temporal Attentive RNN for Node Classification in Temporal Attributed Graphs. In Proceedings of The 29th International Joint Conference on Artificial Intelligence (IJCAI'19)

===>comments after revision =====
concerns are well addressed.

---

> ### Author Response · Authors · 2023-06-29
>
>
> We thank the reviewer for the positive feedback.
> We plan to address the open issues pointed out by the reviewer by implementing the following modifications:
>
> Question: ”While the paper provides a comprehensive overview of existing approaches to temporal GNNs, it does not offer a detailed comparison or evaluation of these approaches.”:
> Answer: We plan to expand the comparison between existing approaches by:
>  - Providing a more comprehensive and precise explanation of each category, accompanied by a mathematical formulation for each one.
>  - Including an additional section titled "Category Comparison" that presents the advantages and disadvantages of each category.
>
> Question: “Some clearly related papers are still missing, such as [...]” :
> Answer: We plan to:
>  - Add a dedicated section “Other approaches to model temporal graphs”, where we intend to introduce and briefly describe the macro categories of techniques used for TG modelling, that are not GNN-based. This section is intended also to better clarify the scope of our survey.
>  - Add a new section “GNN basics”, that is used to recall the definition of the GNN mechanism.
>  - Integrate and discuss accordingly the GNN-methods that were missing from our taxonomy.

---

### Review · Reviewer_m21X · 2023-06-12

**Summary Of Contributions:**

The paper provides a comprehensive survey on graph neural networks (GNNs) for temporal graphs. It introduces a novel formalism for representing temporal graphs that incorporate snapshots and events. The authors define existing learning tasks and specify new learning tasks that currently lack in the temporal GNN (TGNN) community. The paper demonstrates how existing models can be integrated into their framework. Furthermore, the authors discuss a few remaining open challenges in the field of TGNNs.

**Audience:**

Yes

**Broader Impact Concerns:**

I don't have a concern regarding the implications of that work.

**Claims And Evidence:**

Yes

**Requested Changes:**

Please target the weaknesses if possible.

**Strengths And Weaknesses:**

Strengths:
- The paper is well-written and exhibits a clear structure, facilitating readability.
- The formalism introduced for representing data and learning tasks is highly general and well-specified.
- The exploration of unsolved challenges in the field is intriguing and provides valuable insights, making the paper engaging for readers.

Weaknesses:
- While the paper focuses on TGNNs, it lacks explicit specifications of particular models. It would be beneficial to include an overview of how the community has modeled TGNNs and highlight the differences among them.
- The nature of the newly proposed learning tasks, such as temporal node clustering, remains somewhat unclear. It would be beneficial to clarify whether these tasks are simply mathematical extensions within the framework or if there is empirical evidence demonstrating their usefulness. Have domain experts, such as those in the field of epidemics, identified the need for these new learning tasks?
- (Minor) Definition 6 lacks a formal definition of the function T_e(). Although its intended meaning is discernible from the subsequent verbalization, for the sake of clarity, it would be advisable to provide a formal specification.

---

> ### Author Response · Authors · 2023-06-29
>
> We thank the reviewer for the positive feedback.
> We plan to address the open issues pointed out by the reviewer by implementing the following modifications:
>
>
> Question:  “While the paper focuses on TGNNs, it lacks explicit specifications of particular models. It would be beneficial to include an overview of how the community has modeled TGNNs and highlight the differences among them”:
> Answer: we plan to expand the comparison between existing approaches by:
>  - Providing a more comprehensive and precise description of each category of our taxonomy, including also a mathematical formalization for each of them.
>  - Including an additional section titled "Category Comparison" that presents the advantages and disadvantages of each category.
>
>
> Question: “The nature of the newly proposed learning tasks, such as temporal node clustering, remains somewhat unclear. It would be beneficial to clarify whether these tasks are simply mathematical extensions within the framework or if there is empirical evidence demonstrating their usefulness. Have domain experts, such as those in the field of epidemics, identified the need for these new learning tasks?”
> Answer: We agree that a reference to actual applications would be of interest. To this end, we plan to revise the entire “Learning Task on temporal graph” section to point, whenever possible, to actual application papers, in order to highlight their practical usefulness.
>
>
> Question: “(Minor) Definition 6 lacks a formal definition of the function T_e(). Although its intended meaning is discernible from the subsequent verbalization, for the sake of clarity, it would be advisable to provide a formal specification.”
> Answer: We will fix Definition 6 according to the suggestion.

---

### Review · Reviewer_wBZW · 2023-06-26

**Summary Of Contributions:**

The manuscript provides a review of temporal graph neural networks. It consists of formalization of learning settings and tasks, taxonomy categorizing existing approaches, and open challenges discussion.

**Audience:**

Yes

**Claims And Evidence:**

No

**Requested Changes:**

As I mentioned above, there has been one review to discuss temporal graph neural networks in TMLR. I read it and it is much better presented and organized in my opinion.

The manuscripts lack a presentation of graph neural network preliminaries, which should be a key part of the review.

The manuscripts are mainly about all kinds of definitions while there is a lack of detailed discussion about the difference between different methods. A table to summarize that information is suggested.

More related works should be cited and discussed.


**Strengths And Weaknesses:**

Strengths

Provide a review of temporal graph neural networks, which is necessary.

Provide a rigorous formalization of learning settings and tasks, which is new.


Weaknesses

The review lacks a comprehensive review of existing studies on temporal graph neural networks. Most of the reviews are definitions (about 5 pages) while I believe the review should talk more about the methods.

The related work discussions are not sufficient and not well organized. I would suggest the authors read a recent review of graph neural networks in TMLR:

Graph Neural Networks Designed for Different Graph Types: A Survey

In my opinion, it is much better presented and organized. In addition, this review also discusses many temporal graph neural networks.

Missing related papers in temporal graph neural networks. There are some important works that should be discussed and cited, such as:

Heterogeneous temporal graph neural network, SDM 2022

Text-enhanced Multi-Granularity Temporal Graph Learning for Event Prediction, ICDM 2022

---

> ### Author Response · Authors · 2023-06-29
>
>
> We thank the reviewer for the positive feedback.
> We plan to address the open issues pointed out by the reviewer by implementing the following modifications:
>
> Question: ”As I mentioned above, there has been one review to discuss temporal graph neural networks in TMLR. I read it and it is much better presented and organized in my opinion.”:
> Answer: We would like to thank the reviewer for pointing out this very recent survey that, while not specifically focusing on temporal GNNs, does discuss both discrete time and continuous time GNNs, and is definitely worth being cited in our paper. We would like to stress that the novelty and relevance of our contribution is in our opinion the fact of formalizing settings (inductive, transductive, past, future) and tasks that can be defined on temporal graphs, framing existing approaches in this taxonomy and identifying missed opportunities and challenges for further research. This said, we plan to improve the presentation of existing solutions, as discussed further down in the answers to the specific questions by the reviewer.
>
>
> Question: ”The manuscripts lack a presentation of graph neural network preliminaries, which should be a key part of the review.”:
> Answer: We agree that a formal introduction to GNN would be an added value for the paper. We are adding a section “GNN basics”, which introduces the basic mechanism underlying GNNs, and provides a corresponding mathematical formulation. This is later used for the formalization of the categories in the taxonomy.
>
> Question: ”The manuscripts are mainly about all kinds of definitions while there is a lack of detailed discussion about the difference between different methods. A table to summarize that information is suggested.”:
> Answer: To address this unbalance, we plan to update the manuscript along several lines:
>  - We will extend the description of the models in Section 6, and add to the taxonomy a number of models that were not present, including those suggested by the reviewers.
>  - We will add, for the entire taxonomy and for each of its categories, a mathematical formulation of the different mechanisms, in order to highlight the fundamental functioning of each group, and their differences.
>  - We will add a section, at the end of the taxonomy, to discuss and compare the different categories.
>
>
> Question: ”More related works should be cited and discussed.”:
> Answer: We plan to:
>  - Add a dedicated section “Other approaches to model temporal graphs”, where we intend to introduce and briefly describe the macro categories of techniques used for TG modelling, that are not GNN-based. This section is intended also to better clarify the scope of our survey.
>  - Add a new section “GNN basics”, that is used to recall the definition of the GNN mechanism.
>  - Integrate and discuss accordingly the GNN-methods that were missing from our taxonomy.

---

### Review · Reviewer_jHbB · 2023-07-08

**Summary Of Contributions:**

This paper is a comprehensive review of temporal graph neural networks, including a novel formulation for representing temporal graphs using snapshots and events. The authors not only discuss existing learning tasks but also introduce new ones to address current gaps in the TGNN community, and also discuss open challenges.

**Audience:**

Yes

**Claims And Evidence:**

Yes

**Requested Changes:**

Please see the weaknesses.

**Strengths And Weaknesses:**

Strength
--------------
- Providing a comprehensive review of the temporal graph learning approach as well as discussing open challenges, which are beneficial to the community.
- The paper is well-written and easy to follow.
- Presenting and formulating new tasks and temporal graph representations.

Weaknesses
--------------
- There are several important missing works e.g., [1, 2].
- The current version only discusses simple temporal networks. As a comprehensive review paper, it would be beneficial if the authors add even a short discussion about other types of graphs: e.g., hypergraphs, heterogeneous networks, multiplex networks, etc. There are several papers in each of these categories that provide new insights e.g., [3, 4, 5, 6].
- Anomaly detection is one of the important subtasks in temporal graphs. Currently, there is no discussion about it, and it would be better if the authors can address this task in the paper.
- The importance and the application of some of the newly designed tasks are not clear and it would be great if the authors can discuss the potential applications of these tasks (even if there is no dataset for them). It can help future studies address the lack of datasets in these newly designed tasks.








--------

--------

[1] Neural temporal walks: Motif-aware representation learning on continuous-time dynamic graphs. Jin et al., NeurIPS 2022

[2] Well-conditioned Spectral Transforms for Dynamic Graph Representation. Zhou et al. LOG 2022

[3] THINK: Temporal Hypergraph Hyperbolic Network. Agarwal et al. ICDM 2022

[4] CAT-Walk: Inductive Hypergraph Learning via Set Walks. Behrouz et al. Arxiv 2023

[5] Heterogeneous temporal graph neural network. Fan et al. SIAM Data Mining 2022

[6] Heterogeneous Temporal Graph Transformer. Fan et al. KDD 2021

---

> ### Author Response · Authors · 2023-07-13
>
> We thank the reviewer for the positive feedback. We plan to address the open issues pointed out by the reviewer by implementing the following modifications:
>
> Question: ”There are several important missing works e.g., [1, 2].”:
>
> Answer: We plan to include all the mentioned works in the Section “Other approaches to model temporal graphs”.
>
> Question: ”The current version only discusses simple temporal networks. As a comprehensive review paper, it would be beneficial if the authors add even a short discussion about other types of graphs [...] ”:
>
> Answer: We plan to include a discussion about other types of graph in the Section “Other approaches to model temporal graphs”.
>
> Question: ”Anomaly detection is one of the important subtasks in temporal graphs. Currently, there is no discussion about it, and it would be better if the authors can address this task in the paper.”:
>
> Answer: We intend to incorporate a dedicated subsection within the "Learning Task on Temporal Graph" Section, specifically focusing on the Anomaly Detection task.
>
> Question: ”The importance and the application of some of the newly designed tasks are not clear and it would be great if the authors can discuss the potential applications of these tasks (even if there is no dataset for them). It can help future studies address the lack of datasets in these newly designed tasks.”:
>
> Answer: We plan to revise the entire “Learning Task on temporal graph” section to point, whenever possible, to actual application papers, in order to highlight their practical usefulness.

---

### Decision · Action_Editors · 2023-08-09

**Recommendation:** Accept as is

**Comment:**

The paper provides a review of temporal graph neural networks.  Reviewers originally mentioned the following pros and cons:

Pros:
- Review is comprehensive on temporal graph learning, especially on problem settings and task definitions, as well as open problems, which are beneficial for the community.
- Introduces highly general formalism for representing data and learning tasks.
- Well-written and easy to follow.

Cons:
- Lacks of comprehensive review of "methods" and "models".
- Several important works are not discussed.
- Discussion on more complex graphs and anomaly detection are missing.

The authors addressed the reviewer's concerns and revised the paper, reflecting all reviewer's requests.


**Audience:**

yes

**Claims And Evidence:**

yes